# Short- to Mid-Term Clinical and Radiological Results of Selective Laser Melting Highly Porous Titanium Cup in Primary Total Hip Arthroplasty

**DOI:** 10.3390/jcm13040969

**Published:** 2024-02-08

**Authors:** Filippo Familiari, Alessandro Barone, Marco De Gori, Lorenzo Banci, Michelangelo Palco, Roberto Simonetta, Giorgio Gasparini, Michele Mercurio, Giuseppe Calafiore

**Affiliations:** 1Department of Orthopaedic and Trauma Surgery, Magna Graecia University, 88100 Catanzaro, Italy; filippofamiliari@unicz.it (F.F.); alessandro.barone28@gmail.com (A.B.); gasparini@unicz.it (G.G.); 2Research Center on Musculoskeletal Health, MusculoSkeletalHealth@UMG, Magna Graecia University, 88100 Catanzaro, Italy; 3Ospedale Civile di Soverato, 88068 Soverato, Italy; madegori@hotmail.it; 4Clinical Department, Permedica Orthopaedics, 23807 Merate, Italy; lorenzo.banci@permedica.it; 5Division of Orthopaedic and Trauma Surgery, Villa del Sole Clinic, 88100 Catanzaro, Italy; michelangelo.palco@gmail.com (M.P.); simonberto@gmail.com (R.S.); 6Clinica Città di Parma, 43123 Parma, Italy; giuseppe@calafiore.net; 7IRCSS Humanitas Research Hospital, 20089 Rozzano, Italy

**Keywords:** selective laser melting (SLM), highly porous titanium, primary total hip arthroplasty, SLM 3D-printed highly porous titanium acetabular cup, Harris hip score (HHS), osseointegration, radiolucency, bone sclerosis, heterotopic ossifications

## Abstract

(1) **Background**: The aim of this study was to evaluate short- to mid-term clinical and radiological results in patients undergoing primary total hip arthroplasty (THA) with the use of a Selective Laser Melting 3D-printed highly porous titanium acetabular cup (Jump System Traser^®^, Permedica Orthopaedics). (2) **Methods**: We conducted a retrospective study and collected prospective data on 125 consecutive patients who underwent primary THA with the use of highly porous titanium cup. Each patient was evaluated preoperatively and postoperatively with a clinical and radiological assessment. (3) **Results**: The final cohort consisted of 104 patients evaluated after a correct value of 52 (38–74) months. The median Harris Hip Score (HHS) significantly improved from 63.7 (16–95.8) preoperatively to 94.8 (38.2–95.8) postoperatively (*p* < 0.001), with higher improvement associated with higher age at surgery (β = 0.22, *p* = 0.025). On postoperative radiographs, the average acetabular cup inclination and anteversion were 46° (30°–57°) and 15° (1°–32°), respectively. All cups radiographically showed signs of osseointegration with no radiolucency observed, or component loosening. (4) **Conclusions**: The use of this highly porous acetabular cup in primary THA achieved excellent clinical, functional, and radiological results at mid-term follow-up. A better clinical recovery can be expected in older patients. The radiological evaluation showed excellent osseointegration of the cup with complete absence of periprosthetic radiolucent lines.

## 1. Introduction

In orthopedics, the need to use metal devices with a highly porous structure arises from the need to bring implants ever closer to the trabecular structure of cancellous bone in order to fill bone defects and provide structural support and optimal biological anchoring with the bone through the osseointegration and regrowth of new bone tissue in them. Uncemented porous implants are one of the most popular strategies in the orthopedic surgeon’s toolbox to achieve viable anchorage to the host bone [1,2,3,4]. Porous cup fixation relies on a high initial stability due to high friction with the bone and secondary stability achieved through bone growth and regrowth [5], which should ensure long-term implant survival [6,7]. In addition, porous implants are a successful strategy in complex acetabular surgery involving bone loss [8].

The first metal trabecular structures used in orthopedics were manufactured using innovative technology and the metal tantalum, a chemical element with atomic number 73, which is found below niobium on the periodic table of elements. Tantalum is a highly biocompatible and corrosion-resistant metal that has been known and used as an implantable material since the 1950s. The Company (Zimmer Biomet, Warsaw, IN, USA) has patented a technological process for the production of trabecular metal (TM), a highly porous tantalum structure with uniformly interconnected three-dimensional cells that are morphologically very similar to the trabeculae of cancellous bone.

The manufacturing process uses a low-density glassy carbon skeleton formed through the pyrolysis of a polymer foam on which pure tantalum is deposited via chemical vapor deposition. The open, interconnected pore lattice, which enables a very high porosity in the structure, and the micro- and nanotopography of the lattice surface allow the ingrowth and growth of bone. Adjacent bone tissue can grow into the porous structure of the trabecular metal (bone ingrowth) and anchor itself directly to the metal structure (osseointegration or bone ingrowth).

Several materials and different technologies have been used by the medical industry to obtain the optimal porous structure to better promote osseointegration and implant stability. Selective laser melting (SLM) represents one of the more advanced additive manufacturing technologies and consists of a 3D printing process of the component through the iterative application, layer by layer, of a metal powder bed melted by a laser beam [9]. SLM is a technology in which metal parts are produced from a bed of metal powders that are melted with a high-power laser (100 to 1500 W) with a concentrated beam (about 100 µm in diameter). This technology uses powders produced through a micronization process. The shape of the powder grains should be as spherical as possible to enable better distribution when applying the coating. The size can vary depending on the results to be achieved, but there is a tendency to produce powders with a constant particle size, with 15 to 45 µm and 20 to 63 µm being the most commonly used. This 3D printing technology allows for the manufacturing of highly complex designs as irregular highly porous structures without continuity of solution over the outer portion of cementless titanium acetabular cups. Several brands of 3D-printed highly porous titanium acetabular components are available today on the market; however, to date, their clinical evidence in primary THA is still limited to short- or mid-term follow-ups with excellent results [10,11]. The aim of this study was to evaluate the short- to mid-term clinical and radiological results of primary cementless THA with the use of a SLM 3D-printed highly porous titanium acetabular cup.

## 2. Materials and Methods

### 2.1. Study Inclusion, Demographics, and Perioperative Management

An observational retrospective study on prospectively collected data was conducted on 125 consecutive patients who underwent primary THA with the use of the same acetabular cup between January 2017 and December 2019. The research was conducted in compliance with the Declaration of Helsinki. Informed consent was obtained from all participants included in the study. 

The inclusion criteria were (1) age over 18 years at operation, (2) suffering from primary hip osteoarthritis, and (3) a minimum follow-up of 3 years.

The exclusion criteria were (1) revision surgery, (2) femoral neck fracture, (3) patients suffering from a pathological or impending fracture from a tumor, (4) significant cognitive impairment, and (5) failure to understand or complete the questionnaires. 

The data gathered included the patient’s age at operation, sex, and diagnosis of the affected hip.

All surgical procedures were performed by one single surgeon (GC) with a lot of experience in hip arthroplasty. All procedures were performed under spinal anesthesia through a direct lateral approach. All patients were placed in the lateral decubitus position, which was carefully checked to ensure the pelvis was perpendicular to the ground. Deep vein thrombosis (DVT) prophylaxis was carried out through the administration of low-molecular-weight heparin [12], antibiotic prophylaxis was administered intravenously as recommended [13], and in the absence of contraindications, either spinal or epidural anesthesia was performed for all procedures. No specific prophylaxis against heterotopic ossification (HO) was performed [14]. Postoperatively, a multimodal analgesia strategy combining an intravenous formulation of acetaminophen (1 g every 12 h for 5 days), an injectable nonsteroidal anti-inflammatory drug (i.e., diclofenac 75 mg every 12 h for 4 days), and an oral opioid (i.e., tapentadol 50 mg every 12 h for 3 days) was used in the absence of specific contraindications to improve postoperative pain and to reduce the consumption of each agent.

### 2.2. Implant Description

The acetabular shell used in the study cohort was an SLM 3D-printed highly porous titanium cup (Jump System Traser^®^, Permedica Orthopaedics, Merate, Italy). The cup features a hemispherical outer profile with polar flattening and 3 holes for additional fixation with cancellous screws. On the bone interface, the cup features a 1 mm thick highly porous portion of cancellous bone-like irregular titanium lattice (Traser^®^), additively manufactured via SLM with Ti6Al4V alloy powder in a one-step process together with the cup (Figure 1). The characteristics of Jump System Traser^®^ include a 70% porosity, which is an optimal compromise to promote bone growth while maintaining sufficient mechanical properties. The pores of the Jump System Traser^®^ cup have more complex irregular shapes that allow for increased, new bone formation, as demonstrated in Fujibayashi’s study [15]. The open pores are all interconnected and allow complete permeability, as Kuhne explained: “The degree of interconnection is more important for new bone formation than the pore size itself” [16]. The average pore size is 500 μm, a size that ensures faster bone growth, as Karageorgiou found in his study [17]. 

Considering the 1 mm oversizing of the cup, and its high friction that confers the implant a significantly superior grip, the surgeon prepared the acetabular cavity with a reamer of the same size as the cup to achieve a 1 mm press fit. As the cup is 1 mm larger than the reamer, the experience of the surgeon was important for the correct impaction and positioning. The acetabular cup was implanted using vitamin E-blended moderately cross-linked UHMWPE (VitalXE^®^) liners (Permedica Orthopaedics) coupled with ceramic femoral heads (Biolox Delta^®^, CeramTec, Plochingen, Germany). A press-fit conical geometry straight stem with bone-sparing characteristics was used in all implants (SL X-Pore Traser^®^, Permedica Orthopaedics, Merate, Italy). Perioperatively, patients received standard antibiotic and antithrombotic prophylaxis, and multimodal analgesia. A fast-track protocol was used for all patients with active and passive ROM exercises of the lower limb having been initiated at anesthesia resolution and weightbearing with crutches having been allowed at day 1 post operation [18]. 

### 2.3. Functional Assessment

Preoperatively and at the last follow-up, each patient was evaluated using the Harris Hip Score (HHS) [19]. The HHS is a disease-specific test used to evaluate hip disability; scores range between 0 and 100 and include evaluations of pain, function, deformity, and motion domains. A total score of <70 was considered a poor result; 70–79, fair; 80–89, good; and 90–100, excellent. To assess the degree in HHS improvement, a recovery rate (RR) was computed utilizing the following formula: [RR = (postoperative value − preoperative value)/postoperative value × 100]. The postoperative evaluation also included the Forgotten Joint Score 12 (FJS-12) [20]. In FJS-12, all items refer to awareness of the artificial joint (hip or knee) during various daily activities. The final FJS score was determined by summing the responses to the items and converting the raw score into a score on a scale of 0 to 100. In the FJS-12, high scores indicate good performance, i.e., a high level of “forgotten” articulation.

All patients were evaluated for intra- and postoperative complications. Preoperative and postoperative patient assessments were performed by physicians who were not directly involved in the surgery but in the clinical care of the patient.

### 2.4. Imaging Assessment

Preoperatively and at the last follow-up, each patient underwent a standard radiographic evaluation of the hip via both anteroposterior pelvic and axial views. On postoperative radiographs, both the inclination and anteversion of the acetabular component were measured as previously described by Widmer [21]. In detail, inclination was measured directly, while anteversion was computed using the inverse sinus function: $$anteversion=\sin^{-1}\frac{short\;axis}{long\;axis}$$

The acetabular cup osseointegration was assessed according to Moore et al. [22]. Moore defined five radiographic signs for evidence of acetabular osseointegration: (1) the absence of radiolucent lines; (2) the presence of superolateral stiffening; (3) medial stress shielding; (4) radial trabeculae; and (5) inferomedial stiffening. Moore’s study found that the absence of radiolucent lines, the presence of superolateral stiffening, and the presence of medial stress shielding were the most sensitive signs of bone growth. 

Radiolucency, osteolysis, and sclerosis around the acetabular component were evaluated according to DeLee and Charnley [23]. Heterotopic ossifications (HOs) were classified according to Brooker et al. [24]. Two musculoskeletal radiologists who were not directly involved in the study performed the measurements twice, using inter- and intra-observer Cohen’s kappa values of >80% in all cases [25]. 

### 2.5. Statistical Analysis

The distribution of the numeric samples was assessed using the Kolmogorov—Smirnov normality test. Based on this preliminary analysis, a nonparametric set was adopted. Continuous variables were presented as the median and interquartile range (IQR; 25th–75th percentiles), and categorical variables were presented as counts and percentages. The Mann—Whitney test, the Wilcoxon test, and the Kruskal—Wallis test were used, when appropriate, to test the significance of differences between or among values. Univariate linear regressions were performed on the whole population to test possible outcome predictors. Explanatory variables included in the analysis were age (continuous), acetabular radiolucency (categorical), acetabular sclerosis (categorical), and heterotopic ossifications (categorical). The postoperative HHS, HHS variation, and postoperative FJS-12 (continuous variables) were treated as outcomes of the variables. 

Post hoc power was calculated by considering the sample size, the observed effect size, and an α-value of 0.05; a post hoc power greater than 80% was considered appropriate. IBM SPSS Statistics software (version 26, IBM Corp., Armonk, NY, USA) and G*Power (version 3.1.9.2, Institut für Experimentelle Psychologie, Heinrich Heine Universität, Düsseldorf, Germany) were used to construct the database and perform statistical analyses. A *p* value of less than 0.05 was considered significant.

## 3. Results

Out of the initial cohort of 125 patients, 11 patients died of causes unrelated to the procedure, and 10 were lost by follow-up, leaving 104 patients (104 hips) available for complete clinical and radiographic evaluations. The demographic characteristics of the included patients are summarized in Table 1. There were 63 (60.6%) female patients, and the median age at the time of surgery was 73 (65–79) years. All patients suffered from primary hip osteoarthritis. Only in one case, the highly porous acetabular implant was further stabilized with two screws; the patient was an 83-year-old female who suffered from severe osteoporosis. The median follow-up was 61 (52–69) months.

On postoperative radiographs, the average acetabular cup inclination and anteversion were 46° (30°–57°) and 15° (1°–32°), respectively; in 74% and 85% of cases, respectively, the actual values were within the angles of the safe zones as reported by Lewinnek et al. [26]. No patients showed radiolucent lines (Table 2). 

At the last follow-up, none of the patients showed signs of radiolucency. The presence of a superolateral buttress was detected in 95 (91.3%) patients. The presence of a medial stress shield was detected in 85 (81.7%) patients. The presence of radial trabeculae was detected in 70 (67.3%) patients, while the presence of an inferomedial buttress was detected in 75 (72.1%) patients. As a result, at least three signs (3 points) of osseointegration according to Moore et al. [22] were observed in all patients. In detail, 40 (38%) patients showed a very well osseointegrated cup (5 points) (Figure 2), 40 (38%) patients showed a well osseointegrated cup (4 points), and 24 (24%) patients showed an osseointegrated cup (3 points). There were three (2.8%) cases of bone sclerosis around the acetabular component. Heterotopic ossifications occurred in five (4.8%) patients; all cases were classified as grade 1. 

A median postoperative FJS-12 value of 96 (10–100) was recorded, and the median HHS significantly improved from 63.7 (16–95.8) preoperatively to 94.8 (38.2–95.8) postoperatively (*p* < 0.001, 100% power). The median HHS variation with surgery was 26.6 (0–48.5), which is an improvement beyond the minimally clinically important improvement [27,28] in 95.2% of cases.

As shown in Table 3, the occurrence of either sclerosis or heterotopic ossifications around the implant did not significantly affect clinical outcomes.

A statistically significant correlation between postoperative HHS and FJS-12 values was observed (r = 0.838, *p* < 0.001). Regression analyses showed that follow-up time did not affect the postoperative FJS-12 (*p* = 0.597), postoperative HHS (*p* = 0.475), or HHS variation (*p* = 0.493). Despite a lower age at surgery being associated with significantly higher postoperative FJS-12 (β = −0.317, *p* = 0.001) and HHS absolute values (β = −0.317, *p* < 0.001), a higher HHS variation was associated with a higher age at surgery (β = 0.22, *p* = 0.025).

To identify a critical age at operation to potentially predict higher clinical improvement as measured using the HHS variation, arbitrary cutoffs were tested, and the best model (as explained by the lower *p* values detected) has been reported, as previously described [29]. A significant higher HHS variation (*p* < 0.001, 99% power) was found in patients older than 80 years at surgery than in younger individuals. 

## 4. Discussion

The use of a highly porous titanium acetabular cup during primary THA leads to promising clinical and radiological results. This is the first study evaluating the use of the Jump System TRASER^®^ standard acetabular cup in primary THA. 

A previous study by Ciriello et al. [30] explored the use of the Jump System TRASER^®^ dual mobility cup. The results of Ciriello’s study, with a minimum follow-up of 2 years, suggest that dual mobility used in primary THA at high risk of instability could reduce the rate of dislocation.

In the study by Dall’Ava et al. [31], the degree of osseointegration of 3D-printed acetabular cup implants was investigated in comparison with conventionally manufactured porous acetabular cup implants. Bone integration was assessed via macroscopic visual analysis followed by sectioning for non-decalcified histology on eight sections (approximately 200 μm) for each implant. As a result, the area of bone apposition (%), the extent of bone growth (%), the bone-to-implant contact (%), and the depth of growth (%), quantified using the linear intercept method, were considered.

There was no difference in visual bone accrual between the two groups (*p* = 0.209). An analysis of the histologic sections revealed statistically significant differences between the two groups in terms of total BA, extent of ingrowth, BIC, and ingrowth depth (*p* < 0.001), with the 3D-printed group showing higher values. The 3D-printed implants showed a consistently higher bone formation within the porous structure, both in terms of volumetric presence within the available ingrowth space and diffusion (expansion) at the surface of the implants. In the 3D-printed group, bone growth reached the maximum depth more consistently than in the conventional group, and when it did not, the center point reached by the bone in the porous layer was even deeper in the 3D-printed implants.

This emphasizes that 3D-printed implants can promote better osseointegration, which could be due to the ability to create porous structures with optimal morphometric properties that are possible only with 3D printing technology.

Overall, the differences in bone growth between the two groups were not reflected in the clinical outcomes, which were reported to be positive for both uncemented 3D-printed and highly porous acetabular implants produced through conventional means. This suggests that the extent of bone growth, while important, is not the only factor in clinical success. However, the morphometric characteristics of the porous structure, such as porosity and pore size, are key parameters for the implant integration performance. Therefore, 3D printing technology enables better designs of these characteristics by optimizing the amount and position of the metal, creating a predefined optimal porous structure for bone growth.

In our study, we reported a median postoperative HHS of 95 measured at the latest available follow-up. The results of the current study concur with those from previously published studies with the use of SLM-titanium highly porous cups during primary THA. Naziri et al. [11] showed a mean improvement in the HHS from 53 preoperatively to 91 at a mean follow-up of 36 months postoperatively. Geng et al. [10] showed a significant improvement in the HHS from 45 preoperatively to 96 at a mean follow-up of 48 months postoperatively. The long-term study by Huang et al. [32] showed that an excellent HHS of 97 can be expected at a minimum 7-year follow-up, thus indicating that there is no detrimental effect on clinical results overtime. Accordingly, our regression analyses showed that the length of follow-up did not affect postoperative clinical outcomes.

We then observed that a significant reduction in postoperative scores has to be expected with ongoing age. We may justify this result though the observation that HHS values decrease even in the elderly norms [33]. Despite the observation above, we have demonstrated that a better clinical recovery, as measured with the HHS variation with surgery, can be expected in elderly patients. To the best of our knowledge, no previous studies have shown such benefit using porous titanium cups in THA. We can assume that the characteristics of the cup used in this series have to be taken into account to suggest the cup’s use in performing THA in elderly patients and those with a low-quality bone stock.

Radiolucencies around a similar 3D-printed highly porous titanium cup have been reported in 9.1% of cases 2 years after surgery [34], and higher rates of up to 33% have been shown in long-term evaluations [35]. With the use of our SLM-titanium cup, no cases of radiolucencies were found in our series 4 years after surgery, thus suggesting that 3D-printed highly porous titanium acetabular cups might achieve a full osseointegration and contact to the native bone, with a null rate of radiographically evident gaps around the component, at least in the mid-term.

HO prevalence seems to have decreased overtime, likely reflecting the changes in perioperative protocols. Indeed, a recently published systematic review and meta-analysis by Gkiatas et al. [36] reported an overall rate of HO following primary THA of 14% (9–20%) after a mean follow-up of 40 months, with a lower rate of 5% (0–13%) among studies published in the last 15 years. Our data are in full agreement with the results above: at a median follow-up of 52 months, we found a 4.8% rate in HOs. We were able to demonstrate that the occurrence of HOs did not affect our clinical outcomes. We may justify this result through the observation that all cases of HOs in the current study were mild (i.e., grade 1), and it has been previously demonstrated that only severe HOs are of clinical significance [37].

Certain limitations of the current study should be addressed. First, the study design with a single cohort examination might have impeded the testing of the actual value of the used acetabular component in comparison with other implant types. Second, the sample size was too small to generalize our results to a general population of individuals undergoing THA. Indeed, although the sample allowed us to achieve statistically significant and valuable results, our findings should be further confirmed by prospective studies on larger populations. Third, the relatively short length of the minimum follow-up should also be considered. A further limitation is the lack of a quality of life assessment and the lack of comparisons between the immediate postoperative radiographs and those of the last follow-up [38]. However, it should be considered that this series is representative of the first preliminary experiences with this new acetabular cup. The prospective nature of the data collection methods and the use of a validated and standardized functional and radiological assessment represent the considerable strengths of the present study.

## 5. Conclusions

The use of the Jump System TRASER^®^ acetabular cup in primary THA achieves excellent clinical and functional results at mid-term follow-up. A radiological evaluation showed excellent osseointegration of the cup with complete absence of periprosthetic radiolucent lines. Only a few cases of periacetabular HOs, without clinical implications, were observed. Facing our results, a better clinical recovery is to be expected in elderly patients, even more than 80 years old. Surgeons should consider these results when discussing the outcomes of this surgery with patients. Long-term follow-up studies are needed to confirm the longevity of the implant.

## Figures and Tables

**Figure 1 jcm-13-00969-f001:**
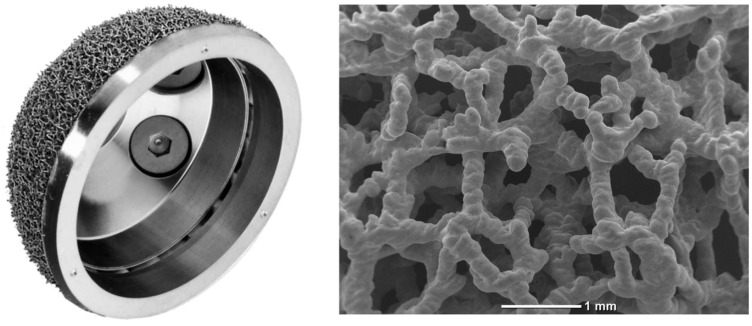
Jump System Traser^®^ highly porous titanium cup. Traser^®^ is an irregular cancellous bone-like highly porous titanium lattice that is 3D-printed via SLM.

**Figure 2 jcm-13-00969-f002:**
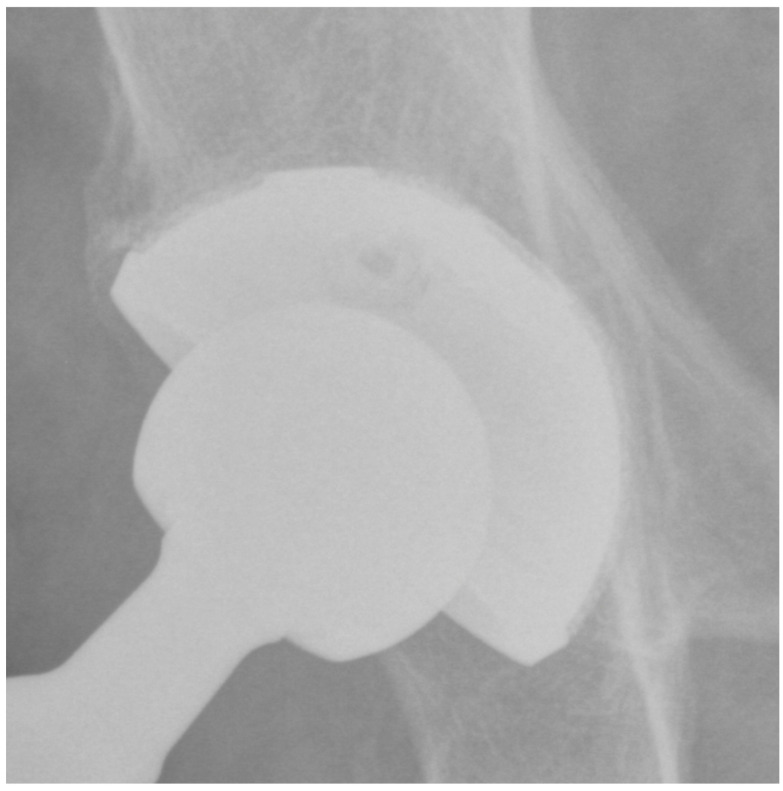
Plain pelvis radiograph of a 79-year-old male patient at 4-year follow-up with 5 points according to Moore.

**Table 1 jcm-13-00969-t001:** Baseline characteristics of included patients (IQR, interquartile range; n, Number of cases).

Patients (n = 104)	Median (IQR) or n (%)
Sex	
Male	41 (39.4%)
Female	63 (60.6%)
Age at surgery (years)	73 (65–79)
Side	
Right	54 (51.9%)
Left	50 (48.1%)
Follow-up (months)	61 (52–69)

**Table 2 jcm-13-00969-t002:** Acetabular cup osseointegration was assessed according to the signs described by Moore et al. [22].

Moore Sign	N (%)
Absence of radiolucent lines	104 (100%)
Presence of a superolateral buttress	95 (91%)
Presence of medial stress-shielding	85 (82%)
Presence of radial trabeculae	70 (67%)
Presence of inferomedial buttress	75 (72%)

**Table 3 jcm-13-00969-t003:** Variation between preoperative and postoperative the Harris Hip Score (HHS) and evaluation with the postoperative Forgotten Joint Score 12 (FJS-12).

		Postop. FJS-12	Postop. HHS	HHS var.
acetabular sclerosis	yes	94 (50–94)	85.8 (74.7–91.8)	28.5 (25.1–30.6)
	no	98 (10–100)	94.8 (38.2–95.8)	26.6 (0–48.5)
		*p* = 0.145	*p* = 0.096	*p* = 0.938
heterotopic ossifications	yes	96 (90–100)	95.8 (85.8–95.8)	30.6 (23.6–47.1)
	no	96 (10–100)	94.7 (38.2–95.8)	26.6 (0–48.5)
		*p* = 0.383	*p* = 0.452	*p* = 0.46

## Data Availability

The data presented in this study are available upon reasonable request to the corresponding author.

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
