# Peer review of "Short- to Mid-Term Clinical and Radiological Results of Selective Laser Melting Highly Porous Titanium Cup in Primary Total Hip Arthroplasty"

_jcm, 2024, doi:10.3390/jcm13040969_

Round 1
Reviewer 1 Report
Comments and Suggestions for Authors
A very interesting study, but it needs more data to validate it.
When I read the abstract, it looked like a very interesting manuscript, but then I had a lot of doubts about how the model was built, as it is not described!
It would be interesting to describe how the implant is constructed. It seems to me that this analysis is purely clinical, but it leaves some doubts as to how the prosthesis is constructed. How was porosity measured, what influence does porosity have on recovery? It would be interesting to carry out a study varying the porosity.
I think this is an interesting preliminary study from a clinical point of view, but more information is needed.

Author Response
Reply: Thanks for your constructive and insightful comments. We updated the manuscript, which is now improved according to your suggestions.
“The acetabular shell used in the study cohort was a SLM 3D-printed highly-porous tita-nium cup (Jump System Traser®, Permedica Orthopaedics, Merate, Italy). The cup features a hemispherical outer profile with polar flattening and 3 holes for additional fixation with cancellous screws. On the bone interface the cup features a 1 mm-thick highly-porous portion of cancellous bone-like irregular titanium lattice (Traser®), addi-tively manufactured by SLM with Ti6Al4V alloy powder in one-step process together with the cup (Figure 1). The characteristics of Jump System Traser® include a 70% porosity, which is an optimal compromise to promote bone growth while maintaining sufficient mechanical properties. The pores of the Jump System Traser® cup have more complex irregular shapes that allow for increased new bone formation, as demonstrated in Fujibayashi's study[15]. The open pores are all interconnected and allow complete permeability, as Kuhne explained: "The degree of interconnection is more important for new bone formation than the pore size itself"[16]. The average pore size is 500 m, a size that ensures faster bone growth, as Karageorgiou found in his study[17].”

Reviewer 2 Report
Comments and Suggestions for Authors
Dear Editor,
Thank you for allowing me to review the manuscript entitled “ Short to Mid-term clinical and radiological results of selective 2 laser melting highly-porous titanium cup in primary total hip 3 arthroplasty” submitted in Journal of Clinical Medicine.
The authors present the results of the evaluation of SLM 3 D printed highly porous titanium acetabular components.
Below some comments:
In introduction, in the second paragraph, I believe it would be beneficial for the readers to have some more information about SLM. Please expand a bit, the rationale behind this technology, how long has it been used, what outcomes have been reported.
The purpose is well written and clear.
In methods section, as the authors stated this cup is 1mm larger than the respective reamer. In all cases you did not ream an additional 1mm. Having some experience with this cup, I find this information important for future studies, because there is a subjective “feeling” that more impact is needed to position this cup. Of course this is not the aim of the study but I think if this info is available it should be stated.
Please also expand on what is standard antibiotic and antithrombotic prophylaxis. Many institutions use different protocols.
At 2.3 : Were there any standard follow-up visits? When the scores were evaluated? Only one time at the end? Please expand.
At 2.4: Pretty much the same comment as 2.3 when the radiologic evaluation took place?
In result, line 146, better rephrase to “ ..11 patients had passed away”
The results are nicely depicted but there is no data regarding when the follow-up occurred? Are all these data at the medial appr. 60 months follow-up??? This should be clear in Methods and Results.
In discussion, line 219 , not “our”, “with the use of SLM-titanium…”
A major limitation that has not been addressed and could be a reason for rejection in some journals is the relatively short period of follow-up. Usually in arthroplasties a follow-up of 5 years is required. In my opinion, this is not a reason to reject this study, but this definitely needs to be stated in the limitations section. Perhaps the authors should consider changing the title (the midterm results is a bit exaggerating- these were mainly short term results)
In general, this is a well-written and interesting paper. Hip surgeons value such data. I congratulate the authors for their work.
Author Response
Reply: Thanks for your constructive and insightful comments. We updated the manuscript, which is now improved according to your suggestions. Please find the detailed point-by point replies below.
- In introduction, in the second paragraph, I believe it would be beneficial for the readers to have some more information about SLM. Please expand a bit, the rationale behind this technology, how long has it been used, what outcomes have been reported.
Reply: Selective laser melting (SLM) represents one of the more advanced additive manufacturing technologies and consists of a 3D-printing process of the component through the iterative application layer by layer of a metal powder bed melted by a laser beam[9]. SLM is a technology in which metal parts are produced from a bed of metal powders that are melted with a high-power laser (100 to 1,500 W) with a concentrated beam (about 100 µm in diameter). This technology uses powders produced by a micronization process. The shape of the powder grains should be as spherical as possible to enable better distribution when applying the coating. The size can vary depending on the results to be achieved, but there is a tendency to produce powders with a constant particle size, with 15 to 45 µm or 20 to 63 µm being the most commonly used. This 3D-printing technology allows to manufacture highly complex designs as irregular highly-porous structures without continuity of solution over the outer portion of cementless titanium acetabular cups. - In methods section, as the authors stated this cup is 1mm larger than the respective reamer. In all cases you did not ream an additional 1mm. Having some experience with this cup, I find this information important for future studies, because there is a subjective “feeling” that more impact is needed to position this cup. Of course this is not the aim of the study but I think if this info is available it should be stated.
Reply: As well described by the reviewer, this cup requires more impact for its placement. The more cups are implanted, the more "feel" the experienced surgeon develops to determine when a TRASER cup is well implanted and stable. As the cup is 1 mm larger than the respective reamer, the experience of the surgeon was important for the correct impaction and positioning. - Please also expand on what is standard antibiotic and antithrombotic prophylaxis. Many institutions use different protocols.
Reply: “Deep vein thrombosis (DVT) prophylaxis was carried out by administration of low-molecular-weight heparin[4], antibiotic prophylaxis was administered intravenously as recommended[5], and in the absence of contraindications, either spinal or epidural anesthesia was performed for all procedures.”
- At 2.3: Were there any standard follow-up visits? When the scores were evaluated? Only one time at the end? Please expand.
Reply: All patients received a follow-up plan on discharge with clinical-functional and radiological (X-ray of the pelvis for hips) checks after 3 months, 6 months, 1 year, 3 years and 5 years. However, the radiographic and clinical-functional evaluations for the study were assessed on the last available follow-up examination (minimum follow-up examination 3 years). We revised the text clarifying that the assessment was performed preoperatively and at the last follow-up. - In result, line 146, better rephrase to “ ..11 patients had passed away”
Reply: “Out of the initial cohort of 125 patients, 11 patients died of causes unrelated to the procedure. - The results are nicely depicted but there is no data regarding when the follow-up occurred? Are all these data at the medial appr. 60 months follow-up??? This should be clear in Methods and Results.
Reply: All patients received a follow-up plan on discharge with clinical-functional and radiological (X-ray of the pelvis for hips) checks after 3 months, 6 months, 1 year, 3 years and 5 years. However, the radiographic and clinical-functional evaluations for the study were assessed on the last available follow-up examination (minimum follow-up examination 3 years). We revised the text clarifying that the assessment was performed preoperatively and at the last follow-up. - A major limitation that has not been addressed and could be a reason for rejection in some journals is the relatively short period of follow-up. Usually in arthroplasties a follow-up of 5 years is required. In my opinion, this is not a reason to reject this study, but this definitely needs to be stated in the limitations section. Perhaps the authors should consider changing the title (the midterm results is a bit exaggerating- these were mainly short term results)
Reply: “Certain limitations of the current study should be addressed. First, the study design with a single cohort examination might have impeded to test the actual value of the used acetabular component in comparison with other implant types. Second, the sample size was too small to generalize our results to the general population of individuals underwent THA. Indeed, although the sample allowed us to achieve statistically significant and valuable results, our findings should be further confirmed by prospective studies on larger populations. Third, the relatively short length of the minimum follow-up should be also considered. A further limitation is the lack of quality-of-life assessment and the lack of comparison between the immediate postoperative radiographs and those of the last follow-up [38]. However, it should be considered that this series is representative of the first preliminary experiences with this new acetabular cup. The prospective nature of the data collection methods, and the use of validated and standardized functional and radiological assessment represent considerable strengths of the present study.”

Reviewer 3 Report
Comments and Suggestions for Authors
Dear Authors,
I read your manuscript with respect and appreciation. Below are some remarks I hope will improve the paper.
1. What is the name of the stem and its manufacturer that was implanted?
2. Did you use a solid back socket only? Were there any cases of implants with holes and additional screw fixation?
3. Were there any problems with postoperative rehabilitation? Lateral approach is usually connected with temporary limping and Trendelenburg’s sign due to irritation of the gluteus medius. Did you notice such problems in operated patients?
4. Crutches are used to decrease weightbearing during the postoperative period, so did your patients really walk with full weightbearing while using crutches?
5. You revealed the improvement in functional results (Harris hip score) by comparing preoperative and postoperative scores. Did you examine patients after standard periods of follow-up (for instance: 3 mths, 6 mths, 12 mths and then annually), or do you just present functional status at the longest possible follow-up?
6. Did you compare the immediate postoperative and long follow-up sockets geometry and presence of radiolucence lines?
7. Why did you not measure the possibility of horizontal and superior migration of the cups by comparing postoperative and long follow-up radiographic measurements?
Author Response
Reply: Thanks for your constructive and insightful comments. We updated the manuscript, which is now improved according to your suggestions.
- What is the name of the stem and its manufacturer that was implanted?Reply: “A press-fit conical geometry straight stem with bone-sparing characteristics was used in all implants (SL X-Pore Traser®, Permedica Orthopaedics, Merate, Italy).”
- Did you use a solid back socket only? Were there any cases of implants with holes and additional screw fixation?
Reply: “Out of the initial cohort of 125 patients, 11 patients were dead For reasons not related to the orthopedic surgery performed, but to comorbidities, and 10 were lost to follow-up, leaving 104 patients (104 hips) available for complete clinical and radiographic evaluation. Only in one case the highly porous acetabular implant was further stabilized with 2 screws; the patient was an 83-year-old female who suffered from severe osteoporosis.” - Were there any problems with postoperative rehabilitation? Lateral approach is usually connected with temporary limping and Trendelenburg’s sign due to irritation of the gluteus medius. Did you notice such problems in operated patients?
Reply: No. We had no cases of Trendelenburg’s sign.
- Crutches are used to decrease weightbearing during the postoperative period, so did your patients really walk with full weightbearing while using crutches?
Reply: We updated the manuscript, which is now improved according to your suggestions. - You revealed the improvement in functional results (Harris hip score) by comparing preoperative and postoperative scores. Did you examine patients after standard periods of follow-up (for instance: 3 mths, 6 mths, 12 mths and then annually), or do you just present functional status at the longest possible follow-up?
Reply: All patients received a follow-up plan on discharge with clinical-functional and radiological (X-ray of the pelvis for hips) checks after 3 months, 6 months, 1 year, 3 years and 5 years. However, the radiographic and clinical-functional evaluations for the study were assessed on the last available follow-up examination (minimum follow-up examination 3 years). We revised the text clarifying that the assessment was performed preoperatively and at the last follow-up. - Did you compare the immediate postoperative and long follow-up sockets geometry and presence of radiolucence lines?
Reply: All patients received a follow-up plan on discharge with clinical-functional and radiological (X-ray of the pelvis for hips) checks after 3 months, 6 months, 1 year, 3 years and 5 years. However, the radiographic and clinical-functional evaluations for the study were assessed on the last available follow-up examination (minimum follow-up examination 3 years). We revised the text clarifying that the assessment was performed preoperatively and at the last follow-up. - Why did you not measure the possibility of horizontal and superior migration of the cups by comparing postoperative and long follow-up radiographic measurements?
Reply: All patients received a follow-up plan on discharge with clinical-functional and radiological (X-ray of the pelvis for hips) checks after 3 months, 6 months, 1 year, 3 years and 5 years. However, the radiographic and clinical-functional evaluations for the study were assessed on the last available follow-up examination (minimum follow-up examination 3 years). We revised the text clarifying that the assessment was performed preoperatively and at the last follow-up. We introduced this topic as limitation of the study. A further limitation is the lack of quality-of-life assessment and the lack of comparison between the immediate postoperative radiographs and those of the last follow-up.

Round 2
Reviewer 1 Report
Comments and Suggestions for Authors
I think is good for publication.